# Effect of the Verbal Encouragement on Psychophysiological and Affective Responses during Small-Sided Games

**DOI:** 10.3390/ijerph17238884

**Published:** 2020-11-29

**Authors:** Hajer Sahli, Okba Selmi, Makrem Zghibi, Lee Hill, Thomas Rosemann, Beat Knechtle, Filipe Manuel Clemente

**Affiliations:** 1Research Unit, Sportive Performance and Physical Rehabilitation, High Institute of Sports and Physical Education of Kef, University of Jendouba, Kef 7100, Tunisia; sahlihajer2005@yahoo.fr (H.S.); okbaselmii@yahoo.fr (O.S.); makwiss@yahoo.fr (M.Z.); 2Division of Gastroenterology & Nutrition, Department of Pediatrics, McMaster University, Hamilton, ON L8S 4L8, Canada; hilll14@mcmaster.ca; 3Institute of Primary Care, University of Zurich, 8091 Zurich, Switzerland; thomas.rosemann@usz.ch; 4Medbase St. Gallen Am Vadianplatz, 9001 St. Gallen, Switzerland; 5Escola Superior Desporto e Lazer, Instituto Politécnico de Viana do Castelo, Rua Escola Industrial e Comercial de Nun’Álvares, 4900-347 Viana do Castelo, Portugal; filipe.clemente5@gmail.com; 6Instituto de Telecomunicações, Delegação da Covilhã, 1049-001 Lisboa, Portugal

**Keywords:** soccer players, encouragement, well-being, recovery state, mood state, enjoyment

## Abstract

Verbal encouragement (VE) is considered as external motivation provided by physical education teacher. For this reason, this study aimed to examine the effects of VE on psychophysiological and affective responses during small-sided games (SSG). Sixteen male school students (age: 17.37 ± 0.48 years) completed four sessions of a 4-a-side SSG. Two sessions occurred with VE (SSGE), and two sessions did not include VE (SSGNE). Heart rate was continuously recorded, and rating of perceived exertion (RPE) and blood lactate concentration ([La]b) were measured after each training session. Physical enjoyment was assessed after each protocol. Mood state was recorded before and after each training session using the profile of mood-state. HR max, [La]b, RPE, Physical enjoyment, and vigor were higher in SSGE compared to SSGNE (all, *p* < 0.001). The SSGE and SSGNE resulted in a decreased total mood disturbance (TMD) (*p* = 0.001, ES = 0.60; *p* = 0.04, ES = 0.33, respectively) and tension (*p* < 0.001, ES = 0.91; *p* = 0.004, ES = 0.47, respectively), and the vigor was increased after the SSGE (*p* < 0.001, ES = 0.76). SSGE and SSGNE induce similar improvement in TMD and tension. However, SSGE induced higher physiological responses, RPE, enjoyment, and positive mood than SSGNE. Physical education teachers could use VE during specific soccer sessions to improve physical aspects, enjoyment, and mood in participants.

## 1. Introduction

During physical education sessions, sports practitioners are dynamically predisposed to improve their physical, cognitive, and emotional capabilities [1], but continuous stimuli provided by the physical education (PE) teacher are needed to make this happen [2]. Thus, during physical activity, PE teachers can stimulate the natural tendencies of active engagement and positive feelings [3]. This stimulation can be created through verbal encouragement, which could influence intrinsic motivation [4]. This, in turn, increases the desire to exercise leading to technical, physical performance, and emotional improvements [4,5,6].

The use of verbal encouragement has been shown to improve motivation and physical performance in various settings and activities, including small-sided games (SSGs) [6,7,8]. SSGs are modified versions of the formal game in which the format of play, pitch dimensions, and rules are constrained to augment the perceptions of players to encourage them to focus on a specific technical action or tactical problem [9,10,11,12].Previous literature has reported that SSGs are an excellent training method for simultaneously improving the physical, physiological, technical, and tactical aspects of soccer players [6,13,14,15].

During integrated training situations, verbal encouragement provided by a coach or a teacher is considered a form of external motivation that positively influences physical engagement, positive behavior, and the desire to train [4,5]. Several studies on SSGs have reported the importance of verbal coach encouragement on the game intensity, as expressed as the perceived exertion (RPE) and physiological responses (i.e., heart rate and lactate concentration) [4,6,9]. For example, Rampinini et al. [7] indicated that during different SSG formats (3 vs. 3, 4 vs. 4, 5 vs. 5, and 6 vs. 6), heart rate (HR), blood lactate concentration ([La]b), and RPE were significantly higher during exercises with encouragement when compared to exercises without encouragement.

Recently, it has been proposed that SSGs are used for affective solicitation [15,16]. In this context, studies have suggested that this training method is more effective pedagogically than other conventional training exercises for reducing the risk of psychological consequences that are associated with a lack of positive affective responses [6,17,18,19]. Indeed, the motivation resulting from SSGs can be more effective in obtaining a positive mood, great physical enjoyment, and high intensity [15,18,20,21]. According to Selmi et al. [6], verbal encouragement leads to increased motivation and physical enjoyment, thus resulting in improved physiological responses during SSGs among soccer players. In addition, Selmi et al. [18] suggested that the motivation resulting from verbal encouragement ensures mood balance during SSGs, while high-intensity intermittent training produces a mood disturbance in soccer players.

Little is known about soccer-specific training regarding the effects of verbal encouragement on players’ performance in civilian team sports. To the best of our knowledge, no studies have addressed the effects of verbal encouragement on psychophysiological and affective responses during SSGs in physical education sessions.

Given the importance of verbal encouragement in athlete motivation, as well as the potential influences of exercise intensity and positive affective responses, research intended to fill this gap in the literature is warranted. Therefore, the aim of this study was to examine the effects of verbal encouragement given by PE teachers during soccer SSGs on the psychophysiological responses, mood state, and physical enjoyment of players. We hypothesized that SSGE would produce higher physiological responses, RPE, enjoyment level, and positive mood than SSGNE.

## 2. Materials and Methods

### 2.1. Ethical Approval

The study was conducted in accordance with the Declaration of Helsinki, and the protocol was fully approved by the research ethics committee of the High Institute of Sports and Physical Education of Kef (ISSEP) (code 2019-0079).

### 2.2. Participants

Sixteen male students from the same secondary school in Tunisia (participants belong to the same study class) were involved in the study (age: 17.37 ± 0.48 years; experience of physical education: 10.7 ± 0.9 years; height: 176.2 ± 5.9 cm; body mass: 68.1 ± 4.1 kg, %Fat: 12.1 ± 2.5%). The inclusion criteria were: (i) All students competed for the same class; (ii) no injury or illness reported one month before the study and during the study; (iii) no physical or cognitive disease reported. the exclusion criteria were: (i) no regular presence of participants in physical education sessions; (ii) participants fell ill during the study period. During this school season, participants trained three days per week (2 sessions’ physical education and 1 session in a school sports club). The participants were familiarized with the experimental protocol and were informed about the procedures. Participants and their parents voluntarily agreed to participate in the study and gave written informed consent after a detailed explanation about the aims and risks involved in the research.

### 2.3. Procedures

This study followed a cross-sectional design. The research was conducted during the 2019–2020 scholar mid-season (eight weeks after the beginning of the season). Before the beginning of the experimental sessions, anthropometric characteristics were assessed, and the students performed the Yo-Yo intermittent recovery test level1 (YYIR-1) to estimate the maximum heart rate (HRmax) [22]. Four sessions of 4-a-side SSG were performed on separate days, each separated by a one-week interval. Each testing protocol (SSG with the physical education teacher’s verbal encouragement (SSGN) and SSG without verbal encouragement (SSGNE)) was repeated twice. During each experimental session, the participants were split into two groups with eight subjects performing SSGE and the other eight performing SSGNE in randomized order. In total, each student performed the SSGE twice, and the SSGNE twice (Figure 1). All measurements were taken on the same synthetic pitch grass at the same time of the day (between 9:00 and 10:30 a.m.) to limit the effects of circadian variations on the measured variables.

HR was continuously monitored during each SSG. Moreover, blood lactate concentration ([La]b), rating of perceived exertion (RPE), and physical activity enjoyment scale (PACES) scores were recorded 5 min after the last bout of the SSG. Participants also had their profile of mood state (POMS) recorded before and after each training session. All participants refrained from strenuous exercise for at least 48 h prior to testing and measurements. Each SSG intervention was preceded by a standardized 15 min warm-up involving low-intensity running, coordination movements, and dynamic stretching and ended with 4 × 8 m sprints; 3 min of recovery separated the warm-up from the first SSG bout [16]. Subjects were allowed to consume drinks during the recovery periods. Subjects were familiarized with the RPE scale, PACES scale, POMS questionnaire, and the SSG regime prior to the beginning of the study. Data were collected by the same sports teacher.

### 2.4. YO-YO Intermittent Recovery Test (YIRT)

The Yo-Yo intermittent recovery test (YIRT) level 1 was completed according to previously described methods [22]. The YIRT is an incremental intensity test used to evaluate aerobic capacity [22]. This protocol consists of repeated 2 × 20 m runs back and forth between the starting, turning, and finishing lines (180° angle), and at a progressively increased speed, which is controlled by audio beeps from a tape recorder. Between each shuttle, the participants have a 10-s active rest period, consisting of 2 × 5 m of walking. The test was performed on a synthetic grass field. The test was stopped when a participant could no longer maintain the required running speed dictated by the beep for two consecutive occasions or felt that he could not complete the stage. The HR was measured and stored using a Polar Team Sport System (Polar-Electro OY, Kempele, Finland). The highest HR average value over 5 s during the test was recorded as YIRT-HRmax. The validity and reliability of this test to determine the aerobic capacity and maximal HR were tested previously [23].

### 2.5. Small-Sided Games

A 4 vs. 4 format (without goalkeepers) played on a 35 × 25 m synthetic pitch (~109 m^2^ per player) was implemented. The session lasted 25 min (four bouts of 4 min separated by 3 min of passive recovery). The SSGE group played the matches while receiving verbal encouragement from the teacher, while the other group (SSGNE) did not receive verbal encouragement. Teacher encouragement involved moving around the perimeter of the field while encouraging the students using soccer-specific terminology and vocabulary (i.e., Go Go Go, Again Again, move, attack the ball, seek the ball, keep the ball, intercept the ball …) and providing new balls when necessary to keep play continuously during the exercise [7,14,19]. The encouragement provided was spontaneous based on the game situation, players’ positioning and movement on the field, space occupation, and ball circulation. During the SSGNE, the physical education teacher stood next to the field and provided new balls when necessary but did not provide any verbal encouragement.

The participants were asked to compete at a maximum effort throughout the exercise and to maintain possession of the ball for the longest time possible. The number of ball touches authorized per individual possession was fixed at two touches to ensure all participants engaged in the SSG.

### 2.6. Physical Activity Enjoyment

The 18-item Physical Activity Enjoyment Scale (PACES) was used to measure positive affect associated with involvement in physical activities in college students [24]. Students were asked to rate “how you feel at the moment about the physical activity you have been doing” using a 7-point bipolar rating scale from 1 (It is very pleasant) to 7 (It is not fun at all). Eleven items are reverse scored. The physical enjoyment score was generated by summing the item scores, which yielded a possible range of 18 through 126. Higher PACES scores reflect greater levels of enjoyment [24]. The PACES scale obtained a Cronbach’s α value of 0.90. Players answered individually the questionnaire.

### 2.7. The Profile of Mood State

Profile of mood states (POMS) [25] questionnaire was used to measure mood disturbance. This self-report questionnaire consists of 65 adjectives designed to assess 6 states (Tension-anxiety, Depression-dejection, Anger-hostility, Vigor-activity, Fatigue-inertia, and Confusion-bewilderment). Responses to each item rated on a 5-point Likert scale (0 indicates “Not at all” and 4 indicates “extremely”). The six subscales of POMS can be combined into a Total Mood Disturbance (TMD) score by summing the *T* scores for the five negative mood subscales and subtracting the *T* score for positive mood state, and adding a constant of 100 in order to prevent negative numbers [TMD = ((Anger + Confusion + Depression + Fatigue + Tension) − Vigor)) + 100]. The Cronbach’s *α* ranged from 0.85 to 0.91. Players answered the questionnaire individually.

### 2.8. Physiological Measures

During the SSG, the HR was continuously monitored using individual HR monitors (Polar Team Sport System, Polar-Electro OY, Kempele, Finland) noted every 5-s intervals. To decrease HR recording error, all students were regularly asked to check their HR monitors throughout the exercise. HR data were, therefore, expressed both as a percentage of HRmax (%HRmax) considering the maximal HR estimated in YYIRT. The mean HR for each bout was calculated to attain the means of the 4 bouts of SSG (HRmean). The %HRmax was calculated by the following formula for each form of SSG, %HRmax: (HRgame/YIRT-HRmax) × 100 [18].

For the determination of [La]b, blood samples were collected 3 min post-training passive recovery [14]. Blood samples, taken from the fingertip of the index finger, were analyzed by a validated portable lactate monitor (Lactate Pro, Arkray, Japan) [14].

### 2.9. Rating of Perceived Exertion

The RPE of each subject was recorded immediately on completion of each session using the 10-point RPE scale proposed by Foster et al. [26] to assess the internal intensity of each SSG intervention. The RPE was measured using a standardized question “How was, and how did you feel the exercise”. This tool has already been used in previous studies [7]. The validity and reliability of this scale to estimate the intensity of effort was also confirmed in previous studies [27]. Players answered individually to avoid hearing the scores of their colleagues. Moreover, players were previously familiarized with the scale to maximize the accuracy of the answers.

### 2.10. Statistical Procedures

All data are expressed as mean ± standard deviation (SD). Before using parametric tests, the assumption of normality was confirmed using the Kolmogorov–Smirnov test. Moreover, the homogeneity of the sample was also tested and confirmed using Levene’s test. Each testing protocol (SSGE and SSGNE) was repeated twice, and the two resulting variables were averaged for analysis. Student’s paired *t*-test was used to compare physiological, physical enjoyment and RPE responses elicited by both filed tests (i.e., [La]b, %HRmax, RPE and PACES scores). The magnitude of change expressed as Cohen’s d coefficient was employed to give a rigorous judgment about the differences between SSGE and SSGNE [28]. The scales of magnitude were considered trivial, small, medium, and large, respectively, for values of 0 to 0.20, >0.20 to 0.50, >0.50 to 0.80, and >0.80 [29]. As for the mood state, a two-way analysis of variance (ANOVA) was used to examine the effect of the “Training session” (SSGE or SSGNE), “Time” (pre-and post-training session), and their interaction (training modality x effort) on the POMS score. When a significant interaction effect was found, the analysis was completed with a post hoc test. Analyses were conducted using the Statistical Package for the Social Sciences (v20.0, SPSS, SPSS Inc., Chicago, IL, USA) and the level of significance was set at *p* < 0.05.

## 3. Results

### 3.1. Physiological Response

Results presented in Table 1 showed significant differences in the %HRmax, [La]b, and RPE variables between SSGE and SSGNE (all, *p* < 0.001).

### 3.2. Physical Enjoyment

The physical enjoyment score was significantly higher (*p* < 0.001, ES = 1.45) in SSGE compared to SSGNE (Figure 2).

### 3.3. Mood State

Concerning the POMS scores, a significant main effect of Time and a significant main effect of the Training session on TMD (Table 2) was observed. Bonferroni post hoc comparisons revealed that the score of TMD decreased significantly for SSGE and SSGNE (Figure 3).

## 4. Discussion

This study assessed the effect of the sports teacher’s verbal encouragement on the physiological responses, mood state, and physical enjoyment of soccer players during SSGs. The main findings of the present study are that (1) SSGE increased physiological and internal intensity to a greater extent than SSGNE, (2) the physical enjoyment was greater after SSGE, and (3) SSGE resulted in a more positive mood state when compared to SSGNE.

The present study showed that the verbal encouragement of a sports teacher positively impacts the physiological responses measured during SSGs. The encouragement variable leads to an increase in the values of %HR max, [La]b, and RPE (4.45%, 22.37%, and 15.10%, respectively). These results suggest that the sports teacher’s encouragement can motivate players to provide a high level of physical engagement, maintain a high work rate during the play situations, and keep possession of the ball for the longest time possible.

Several studies have shown that coach encouragement increases physiological responses and internal intensity, especially in young players [4,6,8,12]. For example, Brandes and Elvers [30] mentioned the effectiveness of verbal encouragement in promoting training intensity and training adherence. This result is also in agreement with those of Rampinini et al. [15], who examined the influence of verbal encouragement on various physiological aspects in several SSG formats (3 vs. 3, 4 vs. 4, 5 vs. 5, and 6 vs. 6) on small, medium-sized, and large pitches. They showed that HR, [La]b, and RPE values were significantly greater in SSGs when verbal coach encouragement was given. Moreover, Selmi et al. [6] compared the effects of coach encouragement during 4 vs. 4 SSGs on physiological responses in youth soccer players. Researchers have also indicated that RPE and %HR max were higher in SSGs with encouragement compared to SSGs without encouragement.

In addition, Edwards et al. [31] reported that verbal encouragement in endurance activities resulted in large improvements in performance and motivation, which has important implications for health, adherence, and physical performance using a practical intervention. These findings suggest that the physiological responses and the internal intensity induced by SSGs might vary according to the motivation of the participants, which can be influenced by a sports teacher or coach.

Overall, the present study results demonstrated that SSGE elicited higher aerobic and anaerobic contributions to energetic demands and perceived exertion, thereby confirming the importance of encouragement in the development of students’ and athletes’ physical fitness.

Physical enjoyment represents a positive affective reaction that allows physical activity to be associated with positive feelings [19,32,33]. Many studies have highlighted the importance of the PACES scale in evaluating physical enjoyment in athletes [19,21,34]. The present study indicates that encouragement has a positive effect on physical enjoyment. The results of the present study are consistent with those reported by Kilit et al. [5], who also indicated that verbal encouragement has a positive effect on participants’ physical enjoyment and commitment to engage in physical exercise. We believe that motivational factors associated with the SSGE condition explain the importance of physical enjoyment. Specifically, participants in this study were motivated by the physical education teacher’s verbal encouragement. Similarly, Midgley et al. [17] indicated that encouragement during training exercises is expected to increase athletes’ motivation and improve positive behavior. In this regard, encouragement during SSG is linked to positive emotional responses to training exercises and is one of the main reasons that motivate participants to contribute to physical activity [4,6,17,31,35].

Another important finding concerns the comparison of mood states (POMS) between the SSGE and SSGNE groups. POMS is commonly used to evaluate the psychological state of participants during physical activity and training [36,37]. Selmi et al. [18] reported that during integrated training, players generally report significant improvements in their mood state. In the present investigation, tension and TMD were positively affected during both SSG conditions (Figure 1). Selmi et al. [18] indicated that high-intensity interval training worsens the mood and is associated with higher scores for anxiety, fatigue, and TMD when compared to SSGs that ensure a mood balance with no change in POMS scores. The stability of the POMS scores in this previous study may be due to the study population (professional soccer players) who participated in the study protocol. Based on the above, it can be suggested that specific soccer training activities decrease negative moods among participants.

Performing SSGE causes a decrease in TMD (as seen in the SSGNE). This indicates that specific training with the ball elicits an increase in positive statements and an increased positive mood. These results are consistent with those presented by Sparkes et al. [38], who examined the influence of SSGs on mood. Decreases in TMD after SSG are usually related to higher vigor and lower tension.

We believe that motivational factors related to the SSGs in the present study may explain the decrease in TMD. Importantly, soccer-specific training leads to mood state improvements. These results suggest that the presence of the ball enhances mood state among participants [6,7,14,18].

The present study indicated that a physical education teacher’s behavior could influence students’ behavior, exercise intensity, affecting psychophysiological and affective responses. Similarly, Jiménez et al. [39] showed that an autocratic teaching style (characterized by negative coach feedback) negatively affected young swimmers’ biological responses, motivational climate, and self-confidence. This kind of teaching style can lead to an affective deterioration. The authors also indicated that swimmers enjoyed training more, were more motivated, and perceived greater personal effort when they were directed by a coach who positively encouraged them. These findings indicate that performance is linked to a positive style of teaching (i.e., teacher’s positive feedback) [39].

Several limitations must be taken into account when interpreting the present results. First of all, the study sample was small due to the difficulty of recruiting a large number of homogeneous participants. Moreover, the testing utilized only one SSG format and a single age cohort of soccer specialist students. Future studies comparing the two SSG conditions should use different levels of determining variables of SSG intensity (i.e., different game rules, duration of each bout, pitch size, number of players, and the presence of goalkeepers) and participants of different ages to extend the applicability of the present findings. Finally, it would be interesting to associate these responses with technical aspects of time-motion parameters (i.e., distance covered, run at high intensity, number of sprints, etc.) because physical aspects are important markers of performance.

This study was conducted in a real training area with youth soccer players, thereby providing practical implications. To the best of our knowledge, this study is the first to compare technical aspects, physiological responses, mood state, and physical enjoyment during SSGs among soccer specialist students. The verbal encouragement of a physical education teacher/coach can be considered an essential factor during specific football training sessions, as it induces physiological, psychological, and technical responses. For this reason, physical education teachers and soccer coaches should verbally encourage their students and players during specific training exercises to increase the game intensity, improve players’ physiological responses, and create positive psychological states.

## 5. Conclusions

SSGE and SSGNE induce similar improvements in total mood disturbance and tension, while SSGE induced more intense physiological responses, internal intensity, enjoyment levels, and positive moods than SSGNE. The results suggest that physical education teachers should increasingly deliver verbal encouragement to enhance the motivation and commitment of students to engage in physical training. Verbal encouragement can be considered as an important variable used in order to improve game performance, physiological responses, and psychological states during specific soccer sessions in school students.

## Figures and Tables

**Figure 1 ijerph-17-08884-f001:**
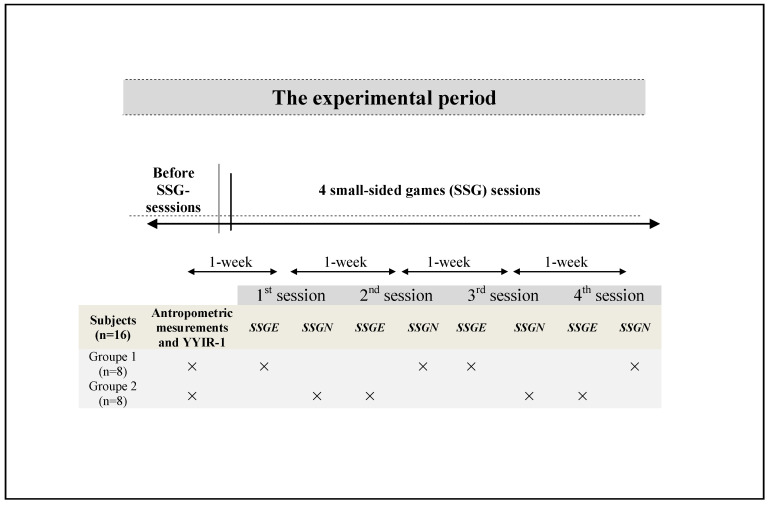
Experimental design figure. SSG: small-sided game, SSGE: SSG with verbal physical education teacher’s encouragement, SSGNE: SSG without verbal physical education teacher’s encouragement, YYIR-1: Yo-Yo intermittent recovery test level 1.

**Figure 2 ijerph-17-08884-f002:**
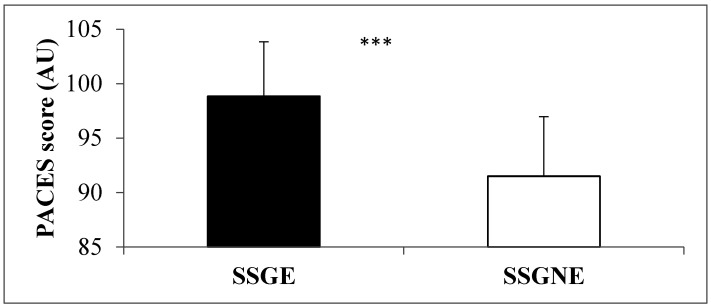
Comparison of perceived physical activity enjoyment between small-sided games with verbal coach encouragement (SSGE) and without verbal coach encouragement (SSGN). *** *p* < 0.001.

**Figure 3 ijerph-17-08884-f003:**
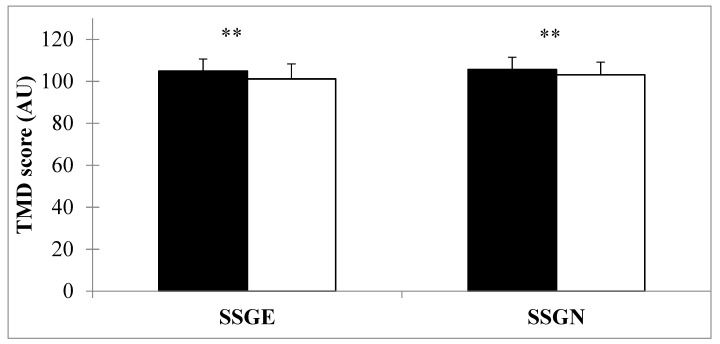
Mean profile of mood states (POMS) score (AU) for both small-sided games training (SSGE) and (SSGNE) collected before (pre-training) and after (post-training) each session. TMD: total mood disturbance. ** *p* < 0.01.

**Table 1 ijerph-17-08884-t001:** Comparison of physiological variables and rating of perceived exertion(RPE) between small-sided games with verbal coach encouragement (SSGE) and small-sided games without verbal coach encouragement (SSGNE).

Variables	SSGE	SSGNE	CI95%	ES	Rating
%HRmax (beat.min^−1^)	88.06 ± 2.25	84.31 ± 2.24 ***	−4.39, −3.10	1.73	Large
[la]b (mmol.l^−1^)	4.43 ± 0.91	3.62 ± 0.6 ***	−1.12, −0.50	1.01	Large
RPE	7.62 ± 0.79	6.62 ± 0.81 ***	0.66, 1.33	1.69	Large

%HRmax: percentage of maximal heart rate; [La]b: blood lactate concentration; RPE: rating of perceived exertion; CI95%, confidence interval of the differences; ES: effect size *** *p*<0.001.

**Table 2 ijerph-17-08884-t002:** Results of the ANOVA with 2 × 2 repeated measures [Training session (SSGE) and (SSGNE)] × Time (pre-and post-test).

Variables	Training Session		Time		Interaction	
	F (1,15)	ES	F (1,15)	ES	F (1,15)	ES
TMD	5.77 *	0.29	62.89 ***	0.80	1.16	0.07

ES: effect size, * *p* < 0.05; *** *p* < 0.001.

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
