# Peer review of "Effect of the Verbal Encouragement on Psychophysiological and Affective Responses during Small-Sided Games"

_ijerph, 2020, doi:10.3390/ijerph17238884_

Round 1

Reviewer 1 Report

That paper is definitely not yet ready.

Conclusions are too short with no key information.

The main problem of that paper "Verbal encouragement (VE) is considered as external motivation" should be rather directed to other Journal. Probably journals from the management area are much more suitable. 

Author Response

Reviewer #1

Comment 1

That paper is definitely not yet ready.

Author’s response: Thank you for the feedback. We do believe that the paper has now. improved significantly

Comment 2

Conclusions are too short with no key information.

Author’s response: Conclusion was improved. Please see in the text

Comment 2

The main problem of that paper "Verbal encouragement (VE) is considered as external motivation" should be rather directed to other Journal. Probably journals from the management area are much more suitable. 

Author’s response: Thank you very much for the suggestion.

Reviewer 2 Report

The manuscript, "Effect of the verbal encouragement on psychophysiological and affective responses during small-sided games" presents a new approach to the collection information of the importance of verbal encouragement in athlete motivation.

The “Introduction” section of the manuscript provide extensive revision and with a very good redaction.  The review of literatura is relevant to their study.

The aim of the study is properly highlighted and justified.

The study followed a cross-sectional design. The presentation of the technique and characterization of the results achieved indicate that the method is quite suitable and in fact could be useful to profundize this aspects in the the social sciences.

Rigorously,  the analisys are  detailed.  The authors do a very good job of presenting a methodology of the accuracy and precision of their results and demonstrate the suitability of the method. Further, the manuscript presents a good and actualized bibliography.  The study is of interest for the scientific community.

Author Response

Reviewer #2

Comment 1

The manuscript, "Effect of the verbal encouragement on psychophysiological and affective responses during small-sided games" presents a new approach to the collection information of the importance of verbal encouragement in athlete motivation.

Author’s response: Thank you for the positive feedback. No changes are required.

Comment 2

The “Introduction” section of the manuscript provides extensive revision and with a very good redaction. The review of literature is relevant to their study.

Author’s response: Thank you for the positive feedback. No changes are required.

Comment 3

The aim of the study is properly highlighted and justified.

Author’s response: Thank you for the positive feedback. No changes are required.

Comment 4

The study followed a cross-sectional design. The presentation of the technique and characterization of the results achieved indicate that the method is quite suitable and in fact could be useful to profundize these aspects in the social sciences.

Author’s response: Thank you for the positive feedback. No changes are required.

Comment 5

Rigorously, the analyses are detailed. The authors do a very good job of presenting a methodology of the accuracy and precision of their results and demonstrate the suitability of the method. Further, the manuscript presents a good and actualized bibliography. The study is of interest for the scientific community.

Author’s response: Thank you for the positive feedback. No changes are required.

Reviewer 3 Report

Accept.

Author Response

Reviewer #3

Comment 1

Accept.

Author’s response: Thank you for the decision. No changes are required.

Reviewer 4 Report

The authors evaluated the effects of verbal encouragement given by PE teachers during soccer games on the psychophysiological responses, mood state, and physical enjoyment of players.

  • The Ethic Committee approval number is missing.
  • How was the study sample calculated? Are sixteen subjects enough to draw statistically significant conclusions?
  • All subjects were males?
  • I recommend to explain better the way in which the subjects were evaluated, that all subjects performed both SSGE and SSGNE and comparisons were made for all subjects (n=18) between SSGE and SSGNE.  See lines 104-107.

Author Response

Reviewer #4

Comment 1

The Ethic Committee approval number is missing.

Author’s response: The Ethic Committee approval number was added

Comment 2

How was the study sample calculated? Are sixteen subjects enough to draw statistically significant conclusions?

Author’s response: Student's paired t-test was used to compare SSGE and SSGNE at the level of physiological, physical enjoyment and RPE responses in 16 participants. In the present study the study sample was mentioned in the limit part.

Comment 3

All subjects were males?

Author’s response: All subjects were males (Sixteen male students)

Comment 4

I recommend to explain better the way in which the subjects were evaluated, that all subjects performed both SSGE and SSGNE and comparisons were made for all subjects (n=18) between SSGE and SSGNE. See lines 104-107.

Author’s response: The explanation of the evaluation procedures of the subjects are well explained ‘’Four sessions of 4-a-side SSG were performed on separate days, each separated by a one-week interval. Each testing protocol [SSG with the physical education teacher’s verbal encouragement (SSGN) and SSG without verbal encouragement (SSGNE)] was repeated twice. During each experimental session, the participants were split into two groups with eight subjects performing SSGE and the other eight performing SSGNE in randomised order. In total, each student performed the SSGE twice and the SSGNE twice (figure 1)’’

  • Each testing protocol (SSGN and SSGNE)] was repeated twice and the two resulting variables were averaged for analysis (the comparison).

Reviewer 5 Report

Comments to authors

Overall, interesting study entitled Effect of the verbal encouragement on psychophysiological and affective responses during small-sided games.

1-            Update references.

2-            Add the characteristics of participants and the exclusion criteria

3-            Figure 1:  vameval test or yo-yo test, please specify the test used in the study.

4-            Please clarify the experimental design and the choice of 4 weeks of training sessions.

Author Response

Reviewer #5

Comment 1

Overall, interesting study entitled Effect of the verbal encouragement on psychophysiological and affective responses during small-sided games.

Author’s response: Thank you for the positive feedback. We do believe that the paper has improved significantly after your valuable suggestions that were addressed accordingly.

Comment 2

Update references.

Author’s response: Thank you for the suggestion. References were updated. Please see in the text and the reference part.

  1. Bujalance-Moreno, P.; Latorre-Román, P. Á.; García-Pinillos, F. A systematic review on small-sided games in football players: Acute and chronic adaptations. J sports sci. 2019,  37, 921-949.
  2. Huhtiniemi, M. ; Salin, K. ; Lahti, J.; Sääkslahti, A.; Tolvanen, A.; Watt, A.; Jaakkola, T. Finnish students’ enjoyment and anxiety levels during fitness testing classes. Phys Edu Sport Ped. 2020 ; 1-15.
  3. Selmi, O.; Ouergui, I.; Castillano, J.; Levitt, D.; Bouassida, A. Effect of an intensified training period on well-being indices, recovery and psychological aspects in professional soccer players. Eur Review Appl Psychol. 2020 ; https://doi.org/10.1016/j.erap.2020.100603.

Comment 2

Add the characteristics of participants and the exclusion criteria

Author’s response: The characteristics of participants and the exclusion criteria were added in the text.

Comment 3

Figure 1: vameval test or yo-yo test, please specify the test used in the study.

Author’s response: Thank you for the comment. Vameval test was chaged to YYIR-1 (Yo-Yo intermittent recovery test level 1).

Comment 4

Please clarify the experimental design and the choice of 4 weeks of training sessions.

Author’s response: Experimental design and the choice of 4 weeks of training sessions were clarified (figure was changed). The explanation of the evaluation procedures of the subjects are well explained ‘’Four sessions of 4-a-side SSG were performed on separate days, each separated by a one-week interval. Each testing protocol [SSG with the physical education teacher’s verbal encouragement (SSGN) and SSG without verbal encouragement (SSGNE)] was repeated twice. During each experimental session, the participants were split into two groups with eight subjects performing SSGE and the other eight performing SSGNE in randomised order. In total, each student performed the SSGE twice and the SSGNE twice (figure 1)’’

  • Each testing protocol (SSGN and SSGNE)] was repeated twice and the two resulting variables were averaged for analysis (the comparison).

Round 2

Reviewer 1 Report

-

Author Response

Reviewer #1

Comment 1

That paper is definitely not yet ready.

Author’s response: Thank you for the feedback. We do believe that the paper has now. improved significantly

Comment 2

Conclusions are too short with no key information.

Author’s response: Conclusion was improved. Please see in the text

Comment 2

The main problem of that paper "Verbal encouragement (VE) is considered as external motivation" should be rather directed to other Journal. Probably journals from the management area are much more suitable. 

Author’s response: Thank you very much for the suggestion. However, verbal encouragement is one of key variables to change acute responses during exercise. Some evidence about that:

Andreacci, J. L., Lemura, L. M., Cohen, S. L., Urbansky, E. A., Chelland, S. A., & Duvillard, S. P. V. (2002). The effects of frequency of encouragement on performance during maximal exercise testing. Journal of sports sciences20(4), 345-352.

Midgley, A. W., Marchant, D. C., & Levy, A. R. (2018). A call to action towards an evidence‐based approach to using verbal encouragement during maximal exercise testing. Clinical physiology and functional imaging38(4), 547-553.

Binboğa, E., Tok, S., Catikkas, F., Guven, S., & Dane, S. (2013). The effects of verbal encouragement and conscientiousness on maximal voluntary contraction of the triceps surae muscle in elite athletes. Journal of sports sciences31(9), 982-988.